# The Mediterranean Diet: From an Environment-Driven Food Culture to an Emerging Medical Prescription

**DOI:** 10.3390/ijerph16060942

**Published:** 2019-03-15

**Authors:** Cristina-Mihaela Lăcătușu, Elena-Daniela Grigorescu, Mariana Floria, Alina Onofriescu, Bogdan-Mircea Mihai

**Affiliations:** 1Diabetes, Nutrition and Metabolic Diseases, “Grigore T. Popa” University of Medicine and Pharmacy, 700115 Iași, Romania; alina.onofriescu@umfiasi.ro (A.O.); bogdan.mihai@umfiasi.ro (B.-M.M.); 2“Sf. Spiridon” Emergency Hospital, 700111 Iași, Romania; floria_mariana@yahoo.com; 3Internal Medicine, “Grigore T. Popa” University of Medicine and Pharmacy, 700115 Iași, Romania

**Keywords:** Mediterranean diet, healthy lifestyle, dietary pattern, chronic diseases, human studies, sustainability, public health nutrition

## Abstract

The Mediterranean diet originates in the food cultures of ancient civilizations which developed around the Mediterranean Basin and is based on the regular consumption of olive oil (as the main source of added fat), plant foods (cereals, fruits, vegetables, legumes, tree nuts, and seeds), the moderate consumption of fish, seafood, and dairy, and low-to-moderate alcohol (mostly red wine) intake, balanced by a comparatively limited use of red meat and other meat products. A few decades ago, the Mediterranean diet drew the attention of medical professionals by proving extended health benefits. The first reports ascertained cardiovascular protection, as multiple large-scale clinical studies, starting with Ancel Keys’ Seven Countries Study, showed a marked reduction of atherosclerotic clinical events in populations with a Mediterranean dietary pattern. Ensuing trials confirmed favorable influences on the risk for metabolic syndrome, obesity, type 2 diabetes mellitus, cancer, and neurodegenerative diseases. While its health benefits are universally recognized today by medical professionals, the present state of the Mediterranean diet is challenged by major difficulties in implementing this protective dietary pattern in other geographical and cultural areas and keeping it alive in traditional Mediterranean territories, also tainted by the unhealthy eating habits brought by worldwide acculturation.

## 1. Introduction

Traditional eating habits seen in geographical territories surrounding the Mediterranean Sea, although differentiated by some food choices and cooking practices specific to each country and culture, share a common set of basic features [1]. The specific dietary dimension of the Mediterranean lifestyle consists of a plant-based cuisine using vegetables, fruits, cereals, nuts, and legumes, most of them cooked by adding substantial amounts of olive oil, with moderate usage of fish, seafood or dairy, and limited intake of meat and alcohol (mostly red wine) [2]. This unique dietary pattern, the result of a complex and multi-millennial interaction between the natural food resources available in the Mediterranean environment and the human element inhabiting the Mediterranean basin throughout history, came to acquire new valences in the last century and to become a precious medical tool in the contemporaneous world [3].

Coinage of the term “Mediterranean diet” and its breakthrough to the attention of the medical public were made possible by the work of Ancel Keys, an American scientist who was the first to notice the relationship between the low incidence of cardiovascular disease in some traditional Mediterranean communities and their specific dietary habits [4]. Ensuing research confirmed the benefits brought by Mediterranean-derived dietary interventions not only in the primary and secondary prevention of cardiovascular disease, but also in the therapeutic approach of obesity, type 2 diabetes, metabolic syndrome, cancer or neurodegenerative diseases [5,6].

In the moment when recognition of the health benefits associated with the Mediterranean diet has become universal, its paradoxical fate is that it is at risk of being extinguished in its homeland territories. Globalization, importation of Western habits, changes in lifestyle and the environment specific to modern civilization have brought a heavy toll on the traditional Mediterranean diet [7]. At the same time, when international guidelines include it among the recommended healthy dietary patterns [8], the United Nations Educational, Scientific and Cultural Organization (UNESCO) considers the Mediterranean diet an “Intangible Cultural Heritage of Urgent Safeguarding” [9]. Given this contradictory stand between universal medical recognition and cultural extinction, this paper aims to review the current information referring to the inception and development of the Mediterranean diet, the major medical evidence supporting its health benefits, and the challenges it must outrun in order to avoid erosion, to maintain survival and sustainability, and to serve public health with the best resources it can offer.

## 2. The Mediterranean Diet: An Environment-Driven Food Culture

The term “Mediterranean diet” is used today to describe the traditional dietary habits of countries neighboring the Mediterranean Sea, mostly Greece and Southern Italy. Nevertheless, it should be understood as more than a strict reference to the preferences these populations exhibit in their daily food selection, since the original meaning of the word *diaita* in Greek does not refer to just food or eating choices, but to a certain “way of living” which corresponds better to the modern concept of “lifestyle” [7].

It is relatively difficult to pinpoint the beginnings of this diet, but it most probably developed along with populations living in the Mediterranean Basin ever since the dawn of civilization. Throughout history, the Mediterranean diet incorporated some of the habits brought by conquerors, while keeping most of the previous local traditions alive and functional. Roots of the Mediterranean diet may be seen in ancient societies belonging to the Fertile Crescent—the Near East geographical area located between the eastern extremity of the Mediterranean Sea and the Persian Gulf, which included Mesopotamia, Canaan and, according to some, Northern Egypt [1].

Foods travelled amongst countries and cultures of the Mediterranean Basin ever since the oldest times. Hieroglyphic records mention exports of wine and olives from Canaan to ancient Egypt. The city-state of Athens depicted an olive tree as its symbol, and the ancient Greeks left the olive branch to humankind as a representation of peace. Greek food influences were brought in the Near East after Alexander the Great conquered this territory in the 4th century BC [10,11,12]. As a plant-based diet, the Mediterranean diet received sequential influences, as successive vegetal species were imported from other geographical regions of the world and acclimatized in the Mediterranean Basin (Table 1).

The food patterns on the shores of the Mediterranean Sea were largely influenced by the three main monotheistic faiths succeeding in this area: Judaism, Christianity, and Islam [7]. These religions also adopted, maintained alive, and held as sacred some of the essential components of the Mediterranean lifestyle [11].

The Mediterranean diet is not, in fact, a unique diet in today’s meaning of the word “diet”. Each of the regions in the Mediterranean Basin developed its own recipes, preferences, and restrictions. The term “Mediterranean diet” could be best understood as a peculiar “dietary pattern” featuring an inter-related set of specific characteristics. Descriptions including only some foods present in the popular culture, while ignoring the absence of other traditional foods or allowing the addition of foods belonging to other eating cultures and patterns should not be accepted as legitimate versions of the Mediterranean diet [14]. An authentic Mediterranean diet pattern should be seen “as a whole”, comprising all its features and not just a part of them [15]. First, olive oil plays a central role in the cooking process, and thus, represents the main source of dietary fat. Cheese is used in limited servings and usually within salads. Meat, milk, and eggs are consumed with a low frequency and in small amounts, and processed meat and sweets are practically non-existent. The Mediterranean diet hence represents, in fact, the only traditional dietary pattern where consumption of saturated and trans fats is inherently minimal. Second, olive oil consumption is associated with a higher vegetable intake, cooked as salads, and to an equally high legume intake in thermic-prepared foods, meaning the Mediterranean diet is essentially a plant-based dietary pattern. Other key components of the Mediterranean diet are the whole grains, nuts, fresh fruits, and a moderate fish intake. Grapes and their derivative products are also used, but one of the main features of the Mediterranean diet is the limited intake of alcohol, as red wine is consumed only with meals, in small servings, with a limited frequency throughout the week, and consumption of other alcoholic beverages such as liquors or beer is not part of the traditional lifestyle [14,15]. However, some variations in food intake between various countries do exist [3]. For example, the total fat consumption largely varies between Greece, where high figures of 40% or more of the total daily caloric intake are reached, and Italy, where fat intake is limited to a moderate consumption of up to 30% of daily calories. The constant feature between different regions of the Mediterranean Basin is rather the high ratio of monounsaturated to saturated fats, which exceeds by far the similar ratios in Northern Europe or Northern America. Differences between countries also occur in the selection of other food sources. The Italian diet features a higher pasta consumption, while the Spanish variant of the Mediterranean diet is characterized by high fish and seafood consumption [3]. A literature review considering the variations amongst the countries with a Mediterranean diet found it to contain, from one case to another, three to nine vegetable servings, half to two fruit servings, one to thirteen cereal servings, and up to eight olive oil servings per day. However, nutrient profiles seem to vary less than the number of different food servings, since in most cases, choices from different food groups complement each other to offer the overall unitary features described above [2].

As a result of such geographical variations in food selection, diverse combinations of food groups are considered by current guidelines to form a Mediterranean diet pattern. Diet pyramids (graphic representations of the main principles within a diet, where foods allowed in larger amounts are represented in the inferior floors of the pyramid and restricted foods are represented towards its top) have today three main variants to describe a Mediterranean diet: the Oldway’s Preservation and Exchange Trust pyramid, the traditional Mediterranean diet of the Greek nutrition guidelines, and the Mediterranean Diet Foundation pyramid. Some of these models kept the features of the traditional food habits, while the others were modified in time in order to better suit nowadays the availability of food supplies, nutritional needs, and eating habits [2].

## 3. Introduction of the Mediterranean Diet into the Medical World

The man responsible for noticing the health protective effects of the Mediterranean lifestyle and for coining the term “Mediterranean diet” is Ancel Keys. A specialist in biology and animal physiology, Keys concentrated at the end of World War II on the effects starvation had on the human body, searching for nutritional techniques able to restore health after starvation [16]. While focusing on starvation, data on morbidity and mortality in post-war Europe came under his eyes. He was surprised to notice the major drop in acute coronary attacks in countries where famine led populations to limit their typical high-fat, high-calorie diets, and also the inverse trend when the same countries recovered after the war and the population feeding changed again. At the same time, Keys was well aware of the high incidence of heart attacks in affluent middle-aged businessmen in the United States, and so he came to suspect that diet may influence health in general and especially the risk for cardiovascular disease. During an era when the concept of risk factors was not yet born, the design of a research study on heart disease in Minnesota businessmen was to become the first prospective study on cardiovascular disease in medical history [4,17,18].

While working in Oxford during a one-year sabbatical in 1951, he came to hear about the very low incidence of heart disease in Southern Italy. Keys went to Naples and opened a portable laboratory there. He was soon able to confirm the previously heard stories about the low incidence of coronary ischemic disease, and he also noticed the low levels of cholesterol most locals exhibited. Keys made similar assessments in other European and African countries, gradually finding that diets rich in saturated fats were associated with increased serum cholesterol levels and a high risk for coronary heart disease [4,19].

When Ancel Keys first presented his ideas of diet causing heart disease at an international meeting of the World Health Organization in 1955, he was met with skepticism and was even challenged by Sir George Pickering, a world-famous cardiologist, to present additional evidence. Unable to do so for the moment, he took this as motivation to design and implement a research project that was to become the so-called Seven Countries Study [18]. He chose to evaluate tobacco use, diet, physical activity, weight status, blood pressure, heart rate, lung capacity, blood cholesterol levels, and electrocardiographic readings in seven cohorts formed from all men aged 40 to 59 inhabiting some well-selected rural communities in the former Yugoslavia, Italy, Greece, Finland, the Netherlands, the United States, and Japan [4]. Yugoslavia was chosen for offering the possibility to study populations with two different eating patterns in coastal and inland regions of the country. Italy was the country where Ancel Keys made his initial observations on the low incidence of heart disease in the setting of a typical (even though not yet designated as so) Mediterranean lifestyle. Greece offered the opportunity to evaluate a population with a high-fat diet, but with a very low intake of saturated fat, as the main source of fats was the monounsaturated fatty acids-rich olive oil. The population of Finland was very well fit but displayed a high incidence of heart disease and a very high intake of saturated fat sources. The Netherlands were representative for a moderate dietary pattern, with a mixed consumption of meat, butter, and vegetables. The United States population sample was chosen both as representative for the high cardiovascular disease incidence and for its geographical stability in time. Japan offered the possibility to study a population with a minimal dietary fat intake. In total, 12,763 subjects were screened. In 5 and respectively 10 years, the study team returned to all of the populations that were initially screened and collected data about the participants who in the meantime experienced a coronary attack [4,18].

When the medical data were submitted to statistical analysis, the results showed significant differences between geographical areas. The lowest rates in heart attack incidences were found in Crete, Japan, and Corfu, in this order. At the other end of the spectrum, the highest rates were identified in Finland, with the United States coming second. Direct comparison between Crete and Finland showed incidences of coronary attacks almost 100-fold higher in the latter (0.1% compared to 9.5%). Seventy-seven percent of the Finish had total cholesterol levels above 200 mg/dL, compared to only 3% of the Japanese. The dietary calories derived from total fat varied between 9% and 40% of the total daily intake, but these figures did not always correlate with the incidence of heart disease, since Greece had one of the highest total fat intakes. Calories deriving from saturated fats varied between 3% and 22%; the correspondence between incidence of heart attacks and saturated fats was convincing [20]. The high intake of saturated fats was thus documented to be associated with a higher incidence of cardiovascular disease in Finland and United States communities [4,18].

Ancel Keys then realized that the dietary habits inherited in traditional Mediterranean populations, especially in Greece and Southern Italy, were associated with a reduced risk of developing cardiovascular disease. He coined these eating habits under the phrase of “Mediterranean diet” and co-authored two books on the subject: *Eat well and stay well* and *How to eat well and stay well the Mediterranean way* [21,22]. He took his own advice on adopting the Mediterranean dietary pattern and died in 2004, at 100 years old, his efforts and research having gained worldwide recognition and respect in the meantime [4].

## 4. Further Confirmation of the Health Benefits of the Mediterranean Diet

The Seven Countries Study had an observational design and limited power to demonstrate a cause–effect relationship. Keys and his team dwelt on the relationship between total serum cholesterol levels and the dietary factors influencing them, more than on the possibility that the Mediterranean dietary pattern as a whole had beneficial effects on cardiovascular health. Even though the Seven Countries Study pointed to the connections existing between eating habits and cardiovascular risk, the concept of the “Mediterranean diet” was kept in the background until the beginning of the 1990s [23].

The Lyon Diet Heart Study was a secondary prevention randomized controlled trial to assess the effects of a modern, French-adapted version of the Mediterranean diet in patients having already suffered from an acute myocardial infarction. In order to best mimic the features of the Greek diet, naturally rich in omega-3 alpha-linolenic acid but poor in omega-6 linoleic acid, the authors decided to use rapeseed oil in association with olive oil. Coming as something of a surprise, the results of this research showed not just a 50% reduction of new acute coronary episodes, but also a reduction in the number of new cancer cases and in all-causes mortality [24,25,26]. The health benefits of the Mediterranean eating style could not be overlooked anymore, and the concept of the “Mediterranean diet” entered the medical consciousness.

In the following years, confirmation of the cardiovascular benefits of the Mediterranean diet became more robust. In the large cohort of the European Prospective Investigation into Cancer and Nutrition (EPIC)–Elderly Prospective Cohort Study, including 74,607 healthy participants from nine European countries, aged 60 or over at the time of recruitment, a variant of the Mediterranean Diet Score was used to estimate the adherence to the Mediterranean diet. The score was obtained by adding nine partial scores of 0 or 1, which represented the intake of nine specific dietary components, and thus varied between a total of 0 (lowest adherence) and 9 (highest adherence). After a 4-year-follow-up, a 2-points increase in the values of this Mediterranean Diet Score was found to be associated with a significant 33% reduction in cardiovascular death [27]. Two other Spanish cohort studies, as well as the multinational Healthy Ageing: a Longitudinal study in Europe (HALE) project, confirmed the association between a higher adherence to the Mediterranean diet and a reduced number of cardiovascular events [28,29,30], also in primary prevention settings. A reduction in the rate of cardiovascular events was also seen in several secondary prevention studies [31,32,33].

One of the recent large trials to provide strong evidence in favor of the Mediterranean diet was the Spanish Prevención con Dieta Mediterránea (PREDIMED) study. Designed as a primary prevention randomized controlled trial, it enrolled 7447 subjects with no clinical signs of cardiovascular disease into either a control group advised to follow a low-fat diet and two active experimental groups set to follow a Mediterranean diet supplemented with extra-virgin olive oil or mixed nuts [34]. Even though all three groups showed a rather small number of acute cardiovascular events, since all of the three diets were healthy cardioprotective eating patterns, the groups randomized to the Mediterranean diet still displayed a 30% reduction in the risk of cardiovascular complications, with an impressive 40% reduction in the risk of stroke [35]. Adherence to the Mediterranean diet was measured in PREDIMED with a dedicated, validated, 14-item screening tool (the Mediterranean Diet Adherence Screener, or MeDiet score) and was found to be inversely associated with the rate of cardiovascular events [36]. Other analyses on the population of the PREDIMED study further showed that the Mediterranean diet seemed to reduce the expression of pro-atherogenic genes [37], cardiovascular risk surrogate markers such as waist-to-hip ratio, lipid fractions, lipoprotein particles, oxidative stress, and markers of inflammation [38,39], but also the risk of developing metabolic syndrome [40] and type 2 diabetes [41]. Nevertheless, the initial results in PREDIMED study were challenged on the account of the small rates of cardiovascular events in all three intervention groups, which could have induced the aforementioned statistically significant differences on the basis of an imperfect randomization procedure, allowing a few biases in the baseline groups’ characteristics [42,43]. The authors chose to retract the first publication [35] and to reanalyze the data after exclusion of sites with randomization deviations; final results still showed significant reductions in the rate of cardiovascular events (31% in the group following a supplementation with extra-virgin olive oil and 28% in the group following a supplementation with mixed nuts) [44].

Attempts to adapt to the Mediterranean eating style and search for related cardiovascular benefits exist today far beyond the borders of the Mediterranean region. Indian patients with pre-existing coronary heart disease or with high cardiovascular risk were included in another randomized trial using a so-called “Indo-Mediterranean diet” rich in whole grains, fruits, vegetables, walnuts, almonds, mustard or soybean oil, all bringing a high content of alpha-linolenic acid, and compared to a control group randomized to a step I National Cholesterol Education Program (NCEP) diet. Patients following the “Indo-Mediterranean diet” had an approximately 60% reduction in the rate of cardiovascular death and an approximately 50% reduction in the risk for non-fatal myocardial infarction [45]. Adherence to the Mediterranean diet was associated with a significantly lower rate of cardiovascular events in a large cohort study following 23,902 UK participants for an average time of 12.2 years [46]; the magnitude of beneficial effects in this study, statistically significant yet inferior to that in PREDIMED, might be attributed to an imperfect, limited transferability of the dietary habits comprised within a typical Mediterranean diet to a British population [47]. Two cohort studies in the United States confirmed that significant reductions in the rate of cardiovascular events were also seen in the American population at higher rates of adherence to the Mediterranean diet [48,49].

At the upper levels of the evidence pyramid, successive meta-analyses of previous cohort studies also acknowledged the association of the Mediterranean diet with reduced rates of cardiovascular morbidity and mortality [50,51,52,53]. A meta-analysis of randomized controlled trials comparing effects of the Mediterranean diet versus low-fat diets on cardiovascular risk factors showed modest but significant benefits of the former on body weight, body mass index, blood pressure, fasting glycaemia, total cholesterol, and high-sensitive C-reactive protein, with no statistically significant differences on low-density lipoprotein (LDL)-cholesterol and high-density lipoprotein (HDL)-cholesterol levels [54]. Another meta-analysis of randomized controlled trials searching for the effects of the Mediterranean-like food patterns in the primary prevention of cardiovascular disease suggested benefits on total and LDL-cholesterol levels [55].

Separate studies confirmed that adherence to the Mediterranean diet was associated with a favorable evolution of abdominal obesity [56], favorable weight changes, and a reduced incidence of overweight and obesity [57]. The protection provided by the Mediterranean diet against the development of type 2 diabetes was confirmed by a systematic review and meta-analysis considering several dietary patterns [58]. From the studies included in this meta-analysis, two prospective clinical trials, one in healthy volunteers [59] and the other in patients with a history of myocardial infarction [60], were specifically designed to evaluate the benefits of the Mediterranean diet in the prevention of type 2 diabetes mellitus. Both these trials found a higher adherence to the Mediterranean diet associated with a reduced risk for developing diabetes [59,60]. A sub-analysis in the EPIC study also described an inverse relationship between the adherence to the Mediterranean diet and the risk of developing diabetes [61]. Studies in patients already diagnosed with type 2 diabetes were fewer, mostly cross-sectional and small-scaled, which may explain why only some of them were able to prove a benefit of the Mediterranean diet on parameters evaluating the glycemic control, while others had neutral results [62,63]. However, no deleterious effects were identified, and benefits in terms of cardiovascular risk reduction were also confirmed in type 2 diabetic patients [63]. Nevertheless, several meta-analyses including clinical trials in already diagnosed type 2 diabetes patients also suggest a beneficial effect of the Mediterranean diet on glycemic control, evaluated by the evolution of plasma glucose and glycated hemoglobin (HbA1c) levels [64,65,66,67].

After studies showing protective effects of the Mediterranean diet against cardiovascular and metabolic diseases, analyses concentrating on possible benefits in other chronic diseases followed. A first indication of a possibly favorable effect of the Mediterranean diet on cancer morbidity and mortality was seen in a secondary analysis of the Lyon Diet Heart Study [25]. Reduced rates of death by cancer were then seen in several studies in Swedish and United States populations [48,68]. According to a recent systematic review and meta-analysis, a higher adherence to the Mediterranean diet seems to have an inverse association with overall cancer mortality and the risk of colorectal, breast, gastric, liver, head and neck, gallbladder and bile ducts cancer [69]. Another focused review suggests a reduced rate for the Mediterranean diet of all digestive cancers, except for pancreatic cancer [70]. The EPIC study is a large-scale, prospective cohort study in 10 European countries, including 521,468 adults followed for a period of 15 years for various cancer, cardiovascular, metabolic, neurodegeneration, and nutrition outcomes; research is still ongoing in multiple working groups and current or future publications are expected to shed light on pathways linking cancer and nutrition [71,72]. For the moment, some components of the Mediterranean diet were suggested to have a strong association with benefits in the primary and secondary prevention of cancer [73].

Some data seem to show a protective effect of the Mediterranean diet against non-alcoholic fatty liver disease, with a higher adherence being associated with a lower severity of hepatic steatosis and reduced levels of alanine–aminotransferase both in cross-sectional and in some low-number, short-term prospective studies [74]. Last but not least, the Mediterranean diet might offer protection against the development of neurodegenerative diseases. In European and United States populations, a better adherence to the Mediterranean diet were found to be associated with a lower risk for cognitive decline and development of Alzheimer’s disease [75,76,77,78]. A large prospective study on 131,368 participants in the American Health Professionals and Nurses’ Health Study showed that higher adherence scores to the Mediterranean diet were associated with a 25% reduction in the risk of developing Parkinson’s disease [79]. According to a 2014 systematic review and meta-analysis, an increased adherence to the Mediterranean diet was associated with a 33% lower risk of mild cognitive impairment or Alzheimer disease and a reduced progression from mild cognitive impairment to clinically overt Alzheimer disease [80].

Previously mentioned data, coming from populations living in India, the United Kingdom, and the United States [45,46,48,49,68,76,77,78,79], are not the only attempts to adapt the Mediterranean diet outside countries in the Mediterranean Basin. A 12-month longitudinal study on healthy Chilean male workers, wherein the Mediterranean diet was implemented in the workplace canteen, obtained improvements in waist circumference, HDL-cholesterol, and blood pressure values, thus reducing the prevalence of the metabolic syndrome by 35% [81]. Another longitudinal 2-year study on obese workers from Israel obtained important weight, triglycerides, and total cholesterol reductions in subjects randomized to a calorie-restricted Mediterranean diet; during an additional 4-year follow-up of the initial subjects, the total weight loss was significantly more important than that obtained by a low-fat, low-calorie diet or by a low-carb diet, thus suggesting these metabolic benefits might originate from a better long-term adherence [82,83]. In a set of studies on United States firefighters, a profession at high risk for cardio-metabolic disease, a greater adherence to the Mediterranean diet was associated with significant improvement of weight, LDL-cholesterol and HDL-cholesterol values, with reductions in total weight, body fat compartment, and prevalence of metabolic syndrome and with higher popularity scores and better adherence between Fire Service members [84]. These data may be justified by the indulgent and appealing lifestyle which is characteristic to the Mediterranean diet, which comprises neither total interdiction in any food group, nor calorie counting [15].

Research attempting to decipher the mechanisms involved in the positive effects of the Mediterranean diet on the risk of cardiometabolic, cognitive or neoplastic diseases covers an increasing number of publications in recent years [5,34,85,86]. Maybe the best way to explain the benefits of the Mediterranean diet is to see it as one of the best illustrations of the concept of “food synergy”, which is a fundamental principle in modern nutrition [87]. Various nutrients and foods present multiple interactions and reciprocally enhance their positive effects, in such a measure that no separate food principle can be taken apart from the context of the whole dietary pattern or be used as an isolated explanation for the benefits brought by the Mediterranean diet altogether. In short, pathways leading to a favorable effect of the Mediterranean diet on various diseases can be systematized as belonging to one or more of the following: lipid lowering and modulating effects; anti-inflammatory, anti-oxidative, and anti-aggregating effects; modulation of cancer-prone mediators such as hormones or growth factors; decreased stimulation of hormonal or other extra- and intracellular transmitting pathways involved in the development of metabolic diseases and cancer, due to the changes in the amino acid content of the diet, compared to other eating styles; changes in gut microbiota, driving a modified production of bacterial metabolites [88]. A sub-analysis in the PREDIMED trial found a higher polyphenol intake to be associated with reduced all-cause mortality; statistically significant differences were seen for stilbenes and lignans, with no significant relationship between flavonoids or phenolic acids and overall mortality [89]. Other data also originating from the PREDIMED trial pointed to the benefits induced by consumption of higher amounts of olive oil in the diet; a 10 g extra-virgin olive oil increase per day was associated with a 10% decrease in the rate of non-fatal cardiovascular events and a 7% decrease in the rate of cardiovascular deaths; rates of cancer and all-cause deaths were not significantly influenced in this report [90]. Olive oil should be understood as more than a vegetal fat comprising predominantly monounsaturated fatty acids such as the oleic acid, but also polyunsaturated fatty acids such as linoleic acids. Since olive oil represents the main source of dietary fat (as intake of milk, butter, cream, cheese or meat are significantly lower in the traditional Mediterranean diet compared to other eating patterns), its use in cooking allows the total amount of saturated fat to be as low as 8%, sometimes throughout the whole life of an individual [88]. The high content in polyphenols and phytochemicals of the olive oil exerts sustained antioxidant actions and reduces the oxidation of unsaturated fatty acids in its composition [87,88]. Moreover, the total antioxidant potential of the Mediterranean diet is completed by the phytochemicals found in whole grains and antioxidant vitamins found in vegetables and fruits. Besides olive oil, the healthy balance of fatty acids in the Mediterranean diet is completed by the polyunsaturated fatty acids brought by the sustained consumption of nuts, seeds, and whole grains and by the moderate or high fish intake. The high content of vegetal fiber brought by the rich consumption of whole grains, legumes, and fruits reduces insulin resistance, inhibits cholesterol absorption in the intestine and cholesterol synthesis in the liver, thus contributing to the overall cardiovascular protection. Phytosterols comprised in nuts, whole grains, seeds, vegetables, and fruits also contribute to the control of the intestinal absorption of cholesterol [88].

A systematic review of experimental studies investigating the relationships between the Mediterranean diet and transcriptomic activity in various tissues found evidence to support this association, although provided by a relatively small number of research papers. Besides the anti-inflammatory actions of the monounsaturated fatty acids found in virgin olive oil, phenolic derivatives such as tyrosol, hydroxytyrosol, secoiridoids, and lignans, also found in olive oil, seem to influence the cell cycle expression, while terpenes such as oleanoic and maslinic acids hold modulatory influences on genes acting on the circadian clock in animal models [91].

## 5. The Mediterranean Diet Nowadays: Between Cultural Erosion and Worldwide Recognition

Like all the other territories of the world, Mediterranean countries were not able to get rid of the current trend of globalization interfering with all cultures, including the one relating to food [92]. Worldwide acculturation is setting a marked stamp on food choices, and exchanges of agricultural products, recipes, and traditions have become a daily rule. As Western food culture, technologies, and advertising are driven by a powerful economic force, they tend to exert a marked influence on traditional eating habits and to substitute them even in their traditional homelands. All of this has resulted in a continuously growing prevalence of excess weight and other eating-related chronic diseases between the last generations of Mediterranean-neighboring populations [93]. Lifestyle standardization, retail sales development, the lesser awareness and appreciation modern generations have for traditional food cultures, which tend to be abandoned in favor of new, socioeconomic-driven changes, women’s integration into the labor market, resulting in limited time for culinary activities, also seem to have a role in the erosion of Mediterranean food cultures [7]. Several surveys of dietary habits performed in Mediterranean regions previously participating in the Seven Countries Study, and featuring low rates of cardiovascular events (Crete, Greece; Nicotera, Crevalcore and Montegiorgio, Italy), showed a decreasing adherence to the Mediterranean dietary traditions manifested by increased intakes of saturated fatty acids, animal foods, cakes, pies, cookies, and sweet beverages, and decreased intakes of monounsaturated fatty acids [94,95]. Most worryingly, low rates of adherence to the Mediterranean diet were seen in multiple studies among children and adolescents in Cyprus and Greece [96,97,98,99].

Environmental difficulties at present are also challenging the sustainability of the Mediterranean way of living. Water scarcity in most Mediterranean-neighboring countries is induced both by decreasing water availabilities and ascending water needs, with agricultural demands representing as high as 64% of the total water expenditure at this moment. Land wasting also has deleterious consequences on food production. Causes of land degradation include expansion of urbanization and associated infrastructures, industrial and urban waste pollution, soil erosion by wind and water, salinization and alkalinization, expansion of tourism-prone littoral zones, sand encroachment, decline of organic matter, all coupled with the limited possibilities of agriculture-matched soil expansion. Global climate changes also have an echo in the Mediterranean Basin, as they induce not only water scarcity and land degradation, but also failure of crops, fisheries, and livestock productions. Last but not least, the biodiversity of species in the Mediterranean regions (previously some of the richest in the world) is continuously declining and a tendency towards monocultures and standardized cultivation practices can be observed, with a negative impact on local food production [7,100].

Given the nutrition challenges occurring today, the pyramid-form graphical representation of the advisable Mediterranean eating pattern had to change in order to adapt to a world where undernutrition coexists with the obesity epidemic induced by a sedentary and hypercaloric lifestyle. The new Mediterranean Diet Foundation Expert Group’s pyramid, for example, introduced the concept of “main meals” in order to emphasize the importance of plant food consumption at each of these meals. Frugality and moderation are also advised at the pyramid’s sides. Other diet-associated elements such as regular physical exercise, adequate rest, the importance of conviviality and culinary activities seen as pleasures and positive occupations, the need for holding in high-esteem local habits, biodiversity, seasonality and the predominant use of traditional, local, and eco-friendly food products are also highlighted [101]. Research aimed at evaluating the combined benefits achieved by associating the Mediterranean eating pattern with systematic induction of weight loss is also under way. A large randomized clinical trial, PREDIMED-PLUS, conducted by the same team of Spanish investigators as PREDIMED and using the same type of Mediterranean-style dietary interventions combined with a three-point weight-loss intervention consisting of restriction of energy intake, physical activity recommendations, and behavioral modifications, finished randomization of 6874 participants in December 2016 and is expected to close in March 2022; design of the study, some baselines, and cross-sectional analyses in PREDIMED-PLUS have recently been published [102,103,104,105,106].

While fighting for sustainability and economic survival in its homelands, the Mediterranean diet must also overcome barriers in other territories of the world, where its health benefits are recognized by the medical community, but adoption by communities is still limited due to the dominance of less healthy Western behaviors. Countries in Northern Europe have started to adopt a Mediterranean-like eating pattern due to the increased availability of Mediterranean fruits and vegetables in local stores and to well-driven public health policies [92]. Acquisition in the United States is still restricted, even though modern nutrition guidelines have already included the Mediterranean eating pattern into their advisable healthy dietary patterns [8]. Paradoxically enough, the Mediterranean diet is not considered anymore a diet for the socially disfavored classes, as it was the moment Ancel Keys made his first scientific observations, but a diet for people with a higher socioeconomic status. However, the truth beneath this concept is not complete. A higher education level is certainly more able to drive people towards learning more about healthy diets, taking into consideration the dietary health advice coming from local and international authorities and finally giving their food choices a higher variability and diversity. However, when speaking strictly about money expenditure, the costs of the Mediterranean diet are close to those of a Westernized diet, because supplementary expenses on fruits and vegetables are counterbalanced by less money spent on red meat, desserts, sweets, and fast foods [107]. A realistic approach to implementing Mediterranean-like eating habits in populations living elsewhere than the Mediterranean Sea coasts could be to search first for local dietary habits by taking an adapted nutritional survey, and then to compare these newly identified eating patterns to the original Mediterranean diet, to identify the major differences and to adapt the local habits to the healthier Mediterranean ones in some key points, without giving up completely on the specific character of local food cultures [108].

## 6. Conclusions

The Mediterranean diet now lies at a crossroads. A product of three millennia of culture and traditions, the Mediterranean lifestyle entered the medical consciousness approximately half a century ago and progressively gained recognition as one of the healthiest patterns of living. Besides cardiovascular, metabolic, cognitive, and possibly anti-neoplastic benefits, the Mediterranean diet seems to be associated with good adherence scores in some extra-Mediterranean populations and with an improved quality of life. Henceforth, it is advised today by a large majority of medical professionals all over the world. At the same time, the erosion of traditions and cultures in the Mediterranean-neighboring populations makes its survival back home an ever more difficult matter. Efforts in these apparently disjunctive directions of both Mediterranean and non-Mediterranean populations are required, in order to make the entire human race benefit from this complex network of food-associated habits that began in times of old as a mixture of lifestyle, religion, and lay culture and which ended up as an emerging medical prescription for health.

## Figures and Tables

**Table 1 ijerph-16-00942-t001:** Geographical origins of plant species included in the Mediterranean diet [3,7,9,13].

**Indigenous plants in the Mediterranean region**	olives, borage, chard, capers, lupines, asparagus, watercress, mallow, thistle, grapes, beet, tiger nut, parsley, cumin, coriander, fennel, oregano, rosemary, sage, lemon balm, savory, fenugreek, bay leaf, saffron, mushrooms
**Plants originating from other Asian regions**	rice, buckwheat, wheat, barley, chickpeas, soybeans, lentils, beans, onions, garlic, leeks, cabbage, broccoli, cauliflower, turnips, spinach, cucumber, yam, arugula, bananas, coconuts, figs, apples, quince, pear, mango, plum, cherry, raspberry, lemon, cucumbers, kiwis, almonds, hazelnuts, walnuts, chestnuts, marjoram, tarragon, pepper, saffron, turmeric, cloves, ginger
**Plants originating from Africa**	millet, sorghum, artichokes, okra, watermelons, melons
**Plants originating from the Americas**	corn, other beans, peanuts, tomatoes, peppers, eggplant, squash, zucchini, potato, sweet potato, prickly pears, cashews, sunflower seeds, avocado, coffee, chocolate, cayenne pepper, allspice, pink pepper

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
