# Peer review of "The Mediterranean Diet: From an Environment-Driven Food Culture to an Emerging Medical Prescription"

_ijerph, 2019, doi:10.3390/ijerph16060942_

Round 1

Reviewer 1 Report

Introduction

·       Line 53. Globalization, import of Western habits, changes in lifestyle and environment specific to modern civilization bring a heavy toll on thetraditional roots the Mediterranean diet is built upon [7].”  Consider change the clause quotes to “the traditional Mediterranean diet.”

2. The Mediterranean Diet: An Environment-driven Food Culture

·       Line 84.  Table 1. I am not an expert in the origins of foods so cannot provided expertise on the accuracy of this table.

·       Line 85. The food patterns on the shores of the Mediterranean Sea were largely influenced by the three main monotheistic faiths succeedinghere in this area, Judaism, Christianity and Islam [7]. These religionsalso adopted, maintained aliveand held as sacred some of the essential components of the Mediterranean lifestyle [11].

·       Line 89.  “Mediterranean diet..”  Should it be “The Mediterranean diet,” here and elsewhere?

·       Line 90.  Each of the regions in the Mediterranean Basin comes withdevelopedits own recipes, preferences and restrictions.

·       Line 92.  Besides its own benefits, olive oil consumption is associated towitha higher vegetableintake, cooked as salads,and to an equallyhigh legume intakein thermic-prepared foods.

3. Introduction of the Mediterranean Diet into the MedicalWorld

·        Line 129.  design of a research study on heart disease in Minnesota

·        Line 140.  by Sir George Pickering, a world-famous cardiologist, to present backing up additional evidence

·        Line 142 .  He chose to evaluate tobacco use, diet, physical activity, weight status, blood pressure, heart frequency, lung capacity, blood cholesterol levels, and electrocardiographic readings in all men aged 40 to 59 inhabiting some provinces in former Yugoslavia, Italy, Greece, Finland, the Netherlands, the United States and Japan [4].

o  ? heart frequency.  Should this be heart rate?

o  Was this study done “in all men” or a sample?  I suspect the latter.

·        Line 155.  “for its geographical stability in time.”  Not sure what this means.  One might think populations in the US may be less stable given in migration.

·            Line 157.  the study team returned to all of the populations in case and collected data about the participants who in the meantime experienced a coronary attack

o    Don’t think “in case” is the correct wording.

·            Line 168.  correspondence with the incidence of heart attackswas closer in their case thecorrespondence between incidence of heartattacksand saturated fats was convincing[18].

Further Confirmation of the Health Benefits of the Mediterranean Diet

·        Line 180.  As such the concept of “Mediterranean diet” was rather disconsidered until the beginning f the 1990s, even though the Seven Countries Study had already changed for good the perception on the connections between eating habits and cardiovascular risk [21].

o  Sentence not clear and needs revision

o  “disconsidered” is not the right word. Maybe something like, “As such the concept of the “Mediterranean Diet” as a healthful diet was questioned…” or something that

o  Line 193. In the following years, confirmation of the cardiovascular benefits of the Mediterranean diet became ubiquitous.

o   “Ubiquitous” is too strong a word.  This paragraph focuses on observational studies and the wording should be consistent with this type of evidence:  suggestive, but not definitive.

·        Line 205.  One of the recent large trials tobringevidence provide strong evidence in favor of the Mediterranean diet is the Spanish PREDIMED study.

·        Line 220.  the groups adoptingrandomized tothe Mediterranean

·        Line 243. At the upper levels of the evidence pyramid

·        Line 271.  Secondary to proof of protective effects of the Mediterranean diet against cardiovascular and metabolic diseases, further analyses followed.

o  Not clear.  Please reword.

·        Line 281.  For the moment, some components of the Mediterranean diet were proven to offer benefits in the primary and secondary prevention of cancer [71].

o  Again, for observational studies and in studies when cancer rates were the not the primary outcome, I think “proven” is too strong a word to use.  Consider “there was a strong association” or something like that.

·        Line 300.  publications inthelast  recent years

The Mediterranean Diet Nowadays: BetweenCultural Erosion and Worldwide Recognition

·        Line 345.  Expanding urbanization and associated infrastructures, industrialand urban waste pollution, wind and water erosion, salinization and alkalinisation, expansion of tourism-prone littoralzones, sand encroachment, decline of organic matter, coupled with the limited possibilities of agriculture-matched soil expansion, gradually lead to severe land degradation, with deleterious consequences on food production

o  Long sentence, consider making it 2

o  “wind and water erosion” not clear

·        Line 370.  consisting in of restriction of energy intake

·        Line 372.  some baseline and cross-sectional analyses in PREDIMED-PLUS have recently began to bebeen published [95–99].

·        Line 376.  but adoption by communities is still partiallylimiteddue to the dominanceof less healthy Western behaviors.

Conclusions

·        Line 405.   as an successful and harmlessemerging medical prescription for health.

Author Response

Reviewer 1:

We   are grateful to the reviewer for the helpful suggestions. We applied all   needed modifications in the text and we think this consistently helped the   manuscript to acquire better clarity and logic.

1. Introduction

 • Line 53. Globalization, import of Western   habits, changes in lifestyle and environment specific to modern civilization   bring a heavy toll on the “traditional roots the Mediterranean diet is built   upon [7].” Consider change the clause quotes to “the traditional   Mediterranean diet.”

The   suggested modification was introduced in the text.

2. The Mediterranean Diet: An   Environment-driven Food Culture

•   Line 84. Table 1. I am not an expert in the origins of foods so cannot   provided expertise on the accuracy of this table.

•   Line 85. The food patterns on the shores of the Mediterranean Sea were   largely influenced by the three main monotheistic faiths succeeding in this   area, Judaism, Christianity and Islam [7]. These religions also adopted,   maintained and held as sacred some of the essential components of the   Mediterranean lifestyle [11].

•   Line 89. “Mediterranean diet…” should it be “The Mediterranean diet,” here   and elsewhere?

•   Line 90. Each of the regions in the Mediterranean Basin developed its own   recipes, preferences and restrictions.

•   Line 92. Besides its own benefits, olive oil consumption is associated with a   higher vegetable intake, cooked as salads, and to an equally high legume   intake in thermic prepared foods.

All   suggested modifications were introduced in the text.

3. Introduction of the Mediterranean   Diet into the Medical World

•   Line 129. design of a research study on heart disease in Minnesota

•   Line 140. by Sir George Pickering, a world-famous cardiologist, to present   additional evidence

•   Line 142. He chose to evaluate tobacco use, diet, physical activity, weight   status, blood pressure, heart frequency, lung capacity, blood cholesterol   levels, and electrocardiographic readings in all men aged 40 to 59 inhabiting   some provinces in former Yugoslavia, Italy, Greece, Finland, the Netherlands,   the United States and Japan [4].

o   ? heart frequency. Should this be heart rate?

o   Was this study done “in all men” or a sample? I suspect the latter.

•   Line 155. “for its geographical stability in time.”  Not sure what this means. One might think   populations in the US may be less stable given in migration.

•   Line 157. the study team returned to all of the populations in case and   collected data about the participants who in the meantime experienced a   coronary attack

o   Don’t think “in case” is the correct wording.

•   Line 168. the correspondence between incidence of heart attacks and saturated   fats was convincing [18].

All   suggested modifications were introduced in the text.

We   changed “heart frequency” with “heart rate”.

The   Seven Countries Study (SCS) pro­to­col chose to include in the follow-up all   men living in seven specific rural communities and corresponding to the age   criterion, and not just a sample of them. We replaced the word “provinces”   that we initially used, in order to avoid any confusion.

The   moment SCS was designed and put into practice, migration of the populations already   existed in other regions and middle-class professions of the USA, but not in   the region and profession chosen by the investiga­tors (railroad workers in   Midwest). We chose not to mention some of these details in the text in order   to avoid diversion from the main ides.

All   suggested modifications were introduced in the text.

Further Confirmation of the Health   Benefits of the Mediterranean Diet

•   Line 180. As such the concept of “Mediterranean diet” was rather   disconsidered until the beginning of the 1990s, even though the Seven   Countries Study had already changed for good the perception on the   connections between eating habits and cardiovascular risk [21].

o   Sentence not clear and needs revision

o   “disconsidered” is not the right word. Maybe something like, “As such the   concept of the “Mediterranean Diet” as a healthful diet was questioned…” or   something that

•   Line 193. In the following years, confirmation of the cardiovascular benefits   of the Mediterranean diet became ubiquitous.

o   “Ubiquitous” is too strong a word. This paragraph focuses on observational studies   and the wording should be consistent with this type of evidence: suggestive,   but not definitive.

•   Line 205. One of the recent large trials to provide strong evidence in favor   of the Mediterranean diet is the Spanish PREDIMED study.

•   Line 220. the groups randomized to the Mediterranean

•   Line 243. At the upper levels of the evidence pyramid

•   Line 271. Secondary to proof of protective effects of the Mediterranean diet   against cardiovascular and metabolic diseases, further analyses followed.

o   Not clear. Please reword.

•   Line 281. For the moment, some components of the Mediterranean diet were   proven to offer benefits in the primary and secondary prevention of cancer   [71].

o   Again, for observational studies and in studies when cancer rates were the not   the primary outcome, I think “proven” is too strong a word to use. Consider   “there was a strong association” or something like that.

•   Line 300. publications in recent years

The   sentence was revised according to the indications of the reviewer.

All   suggested modifications were introduced in the text.

The   sentence was revised according to the indications of the reviewer.

The   sentence was revised according to the indications of the reviewer.

The   suggested modification was introduced in the text.

The Mediterranean Diet Nowadays:   Between Cultural Erosion and Worldwide Recognition

•   Line 345. Expanding urbanization and associated infrastructures, industrial   and urban waste pollution, wind and water erosion, salinization and alkalinisation,   expansion of tourism-prone littoral zones, sand encroachment, decline of   organic matter, coupled with the limited possibilities of agriculture-matched   soil expansion, gradually lead to severe land degradation, with deleterious   consequences on food production

o   Long sentence, consider making it 2

o   “wind and water erosion” not clear

•   Line 370. consisting of restriction of energy intake

•   Line 372. some baseline and cross-sectional analyses in PREDIMED-PLUS have recently   been published [95–99].

•   Line 376. but adoption by communities is still limited due to the dominance   of less healthy Western behaviors.

The   sentence was revised according to the indications of the reviewer.

All   suggested modifications were introduced in the text.

Conclusions

•   Line 405. as an emerging medical prescription for health.

The   suggested modification was introduced in the text.

Reviewer 2 Report

A well done work and nicely organized. However, in my opinion some modifications are required to avoid inaccurate and misleading conclusions. 

I believe it is best to modify the title of the paper itself. Mediterranean Diet is not an endangered food culture and also it is not beneficial only regarding non-communicable diseases. It has been found to reduce all-cause mortality, improve overall quality of life, etc.. Same idea holds for the conclusions of the authors that need to be modified accordingly! 

For this, authors need to read relevant papers from the group of Korre et al. Some of them are listed below. 

Korre M, Sotos-Prieto M, Kales SN. Survival Mediterranean Style: Lifestyle Changes to Improve the Health of the US Fire Service. Front. Public Health.2017; 5:331.doi: 10.3389/fpubh.2017.00331

 Korre M, Kalogerakou T, Sotos Prieto M, Kales SN, What is the Mediterranean Diet and How Can It Be Used to Promote Workplace Health? Journal of Occupational and Environmental Medicine Forum.2016; 58:3.

Author Response

Reviewer 2:

I   believe it is best to modify the title of the paper itself. Mediterranean   Diet is not an endangered food culture and also it is not beneficial only   regarding non-communicable diseases. It has been found to reduce all-cause   mortality, improve overall quality of life, etc. Same idea holds for the   conclusions of the authors that need to be modified accordingly!

For   this, authors need to read relevant papers from the group of Korre et al.   Some of them are listed below.

Korre   M, Sotos-Prieto M, Kales SN. Survival Mediterranean Style: Lifestyle Changes   to Improve the Health of the US Fire Service. Front. Public Health.2017;   5:331.doi: 10.3389/fpubh.2017.00331

Korre   M, Kalogerakou T, Sotos Prieto M, Kales SN, What is the Mediterranean Diet   and How Can It Be Used to Promote Workplace Health? Journal of Occupational   and Environmental Medicine Forum.2016; 58:3.

We   thank the reviewer for pointing out these sources to us. We found them   relevant and very helpful, so we incorporated this information in the paper.   We modified the title and the con­clusions accordingly.

Reviewer 3 Report

This manuscript is the informative report to show “Mediterranean Diet: A Solution for Non-Communicable Diseases Provided by an Endangered Food Culture”. Using various studies of original research, cohort, systematic review and Meta-analysis, this review was good according to your hypothesis, but more informative discussion for following doubtful points are recommended.

  Mediterranean diet (Med diet) is a well-known diet with many health benefits on non-communicable diseases (NCD) such as CVD, DM, cancer, dementia, and so on. However, the biggest stumbling block of defining the Med diet as a solution of NCD is that Med diet pattern should be universally admitted to geographically distinct or racially different countries beyond the Mediterranean sea. You need to refer to other intervention studies of Med diet on different regions (territories) and races and provide universal evidence among them.

  Also, your argument would be much stronger with proofs that precisely show the opposite pattern in countries with high incidence of CVD such as Finland, USA, and others, compared to the Med diet. Your manuscript focused on the high ratio of monounsaturated (MUFA) to saturated fatty acids (SFA), low calories and low fat (<30%). That fat intake less than 30% is the most important factor to reduce CVD prevalence is well known, although, CVD prevalence is growing in some populations even in low fat consumers such as Asians. Therefore, I do agree that the component of fat is more important in this issue. However, the actual component that is well known to significantly reduce CVD is PUFA, in particular w-3, rather than MUFA; olive oil which consists w-9. Could you propose any intervention study that w-9 has at least equal to better outcomes in reducing CVD risks compared to  w-3, which can further strengthen your hypothesis?

  You must first clarify what the “centerpiece” of Med diet to reduce CVD and to overcome genetic differences (=different races) is. Then, you should mention what exact component is modified from the typical Med diet amongst all the “modified Med diets” appeared in the manuscript. Lastly, you should provide what outcomes differ in how much/many amongst all the “modified” Med diets appeared in the manuscript, compared to your suggested typical Med diet.

For example, you have already mentioned that “olive oil” plays a central role in a typical Med diet in ‘page 3, line 91~92’. If the Spanish or Japanese diet, which consists more fish (w-3) than olive oil (w-9), has an evidence of CVD reduction AND if you refer those diet patterns as “one of modified Med diets”, then the typical food selection in the Med diet and its pattern loses its superiority as well as its evidential potentials. Also, you mentioned that typical Med diet consists plant-based cuisine of olive culture. However, the diet also includes dairy products, meats, and alcohol which are as well found in other dietary patterns. You should specifically point out the major/core diet that can distinctively discriminate the Med diet from the other patterns, and you should NOT refer diets that have modified the core food as “the modified Med diet”.

Moreover, the crude ratio of PUFA to SFA throughout the population could NOT be easily changed by diet modification, since the diet pattern is built on various environmental circumferences for a long time such as cultures (history), lifestyle (living pattern), in particular genetic imprints and the effects. Therefore, the typical Med diet, including the modified ones, might be an ideal choice to reduce CVD in certain populations but it cannot be a one-in-all solution for current/future CVD patients in different countries/races, unless you provide scientific evidence on this matter.

  Finally, although you did not specifically describe in this manuscript, you did mention “red wine” as one of important components of the Med diet (page 1, line 38). The evidence of CVD risk reduction holds for resveratrol itself but not for resveratrol + alcohol combination. Therefore, if your intention was to provide the whole Med diet as a solution, there should be an explanation of either 1) the benefit of consuming red wine with scientific evidence or 2) yet debatable benefits of wine consumption with no recommendation at all.

Author Response

Reviewer 3:

We   thank the reviewer for pin­poin­ting some impor­tant issues in the content of   our manuscript. Please find below the answers to each of these remarks and   the description of the modifications that were intro­duced in the text. We   think the fragments we changed based on the comments of our reviewer helped   the text to increase its clarity and value.

1.   Mediterranean diet (Med diet) is a well-known diet with many health benefits   on non-communicable diseases (NCD) such as CVD, DM, cancer, dementia, and so   on. However, the biggest stumbling block of defining the Med diet as a   solution of NCD is that Med diet pattern should be universally admitted to   geographically distinct or racially different countries beyond the   Mediterranean sea. You need to refer to other intervention studies of Med   diet on different regions (territories) and races and provide universal   evidence among them.

Since   the initial version of our paper included limited referrals to the im­ple­men­ta­tion   of the Mediterranean diet in different regions (territories) and races, we extended   the information and added supplementary descriptions of studies using a   Mediterranean diet intervention in such a context.

2.   Also, your argument would be much stronger with proofs that precisely show   the opposite pattern in countries with high incidence of CVD such as Finland,   USA, and others, compared to the Med diet. Your manuscript focused on the   high ratio of monounsaturated (MUFA) to saturated fatty acids (SFA), low   calories and low fat (<30%). That fat intake less than 30% is the most   important factor to reduce CVD prevalence is well known, although, CVD   prevalence is growing in some populations even in low fat consumers such as   Asians. Therefore, I do agree that the component of fat is more important in   this issue. However, the actual component that is well known to significantly   reduce CVD is PUFA, in particular w-3, rather than MUFA; olive oil which   consists w-9. Could you propose any intervention study that w-9 has at least   equal to better outcomes in reducing CVD risks compared to w-3, which can   further strengthen your hypothesis?

We   introduced descriptions of dietary patterns distinct from the Mediterranean   diet existing in countries with higher incidence of CVD in the text.

Publications   explaining the health benefits of the Mediterranean diet usually describe   this dietary pattern as “a whole” and justify its positive effects by a   so-called “food synergy” where each nutrient and food positively influence   the effects of others; hence, no nutrient can be taken out of context to   explain a specific health benefit brought by the whole diet. This is also the   case of the oleic acid, which is not the only significant constituent of the   olive oil or of the Mediterranean dietary pattern. Therefore, we changed our   text accordingly, by introducing specifications about this concept of “food   synergy” underlying the positive effects of the Mediterranean diet and by   extending the section describing its potentially beneficial multiple   mechanisms and inter-relations existing between foods and nutrients.

3.   You must first clarify what the “centerpiece” of Med diet to reduce CVD and   to overcome genetic differences (=different races) is. Then, you should   mention what exact component is modified from the typical Med diet amongst   all the “modified Med diets” appeared in the manuscript. Lastly, you should   provide what outcomes differ in how much/many amongst all the “modified” Med   diets appeared in the manuscript, compared to your suggested typical Med diet.  

For   example, you have already mentioned that “olive oil” plays a central role in   a typical Med diet in ‘page 3, line 91~92’. If the Spanish or Japanese diet,   which consists more fish (w-3) than olive oil (w-9), has an evidence of CVD   reduction AND if you refer those diet patterns as “one of modified Med   diets”, then the typical food selection in the Med diet and its pattern loses   its superiority as well as its evidential potentials. Also, you mentioned   that typical Med diet consists in plant-based cuisine of olive culture.   However, the diet also includes dairy products, meats, and alcohol which are   as well found in other dietary patterns. You should specifically point out   the major/core diet that can distinctively discriminate the Med diet from the   other patterns, and you should NOT refer diets that have modified the core   food as “the modified Med diet”.

Moreover,   the crude ratio of PUFA to SFA throughout the population could NOT be easily   changed by diet modification, since the diet pattern is built on various   environmental circumferences for a long time such as cultures (history),   lifestyle (living pattern), in particular genetic imprints and the effects.   Therefore, the typical Med diet, including the modified ones, might be an   ideal choice to reduce CVD in certain populations but it cannot be a   one-in-all solution for current/future CVD patients in different   countries/races, unless you provide scientific evidence on this matter.

As   mentioned before, current literature did not choose to describe only one “core”   element of the Mediterranean diet to be responsible for its whole cardiovascular   benefits. Variations between the dietary habits existing in different regions   of the Mediterranean basin do exist, but are always subordinated to the same   set of general features which combine to form the Mediterranean dietary   pattern, so they should not be interpreted as “modified Mediterranean diets”.   In order to make this aspect more evident, we chose to keep the fragment in   the initial text which emphasized that the nutrient composition in the   Mediterranean diet is remarkably stable, even though the number of servings   may apparently vary more from one Mediterranean country to the other. We also   chose to extend fragments describing the main features of the Mediterranean   diet and the importance of them existing altogether within a dietary pattern   in order to be able to call it an authentic Mediterranean diet.

Even   though we initially used the term “modified” in order to designate variants   of the Mediterranean Diet Score (an instrument used to assess adherence to   the Mediterranean diet), and not to designate the Mediterranean diet itself,   we changed this term throughout the text, in order to avoid any confusion.

As   the Spanish diet features a higher consumption of fish than other areas in   the Mediterranean Basin, but also the other components of the traditional   Mediterranean diet, it can be considered a variant of this dietary pattern,   and not a modified Mediterranean diet. This is not the case with the Japanese   diet, which does not include the high intakes of nuts, legumes, vegetables or   whole cereals which are highly specific to the Mediterranean diet, but only a   sustained fish intake. Therefore, the Japanese diet cannot be seen as an   equivalent for the Mediterranean diet, not even for a modified one, since the   concept of “food synergy” we previously described is not possible in its case.

Contrary   to foods that were previously mentioned, consumption of dairy products, meats   and alcohol is typically limited in the traditional Mediterranean diet. It is   exactly the sum of these limitations which constitutes another major   difference from other dietary patterns. We introduced some supplementary   explanations in the text on this matter, in order to make this aspect as evident   as possible and avoid any confusion in our paper.

In   order to emphasize the practical potential of the Mediterranean diet in non-Mediterranean   popula­tions, we also included in the paper the description of a study   performed on some US professionals (firefighters), where this dietary pattern   was inclu­ded in their lifestyle with good results, but also with a high   acceptance and adherence.

4.   Finally, although you did not specifically describe in this manuscript, you   did mention “red wine” as one of important components of the Med diet (page   1, line 38). The evidence of CVD risk reduction holds for resveratrol itself   but not for resveratrol + alcohol combination. Therefore, if your intention   was to provide the whole Med diet as a solution, there should be an   explanation of either 1) the benefit of consuming red wine with scientific   evidence or 2) yet debatable benefits of wine consumption with no   recommendation at all.

As   alcohol intake is concerned, the main feature of the Mediterranean diet is   the total avoidance of spi­rits or beer, and a consistent limitation in the   red wine intake, which is traditionally consumed in a reduced number of   servings, only with meals and at a low frequency throughout the week, without   a pattern of binge drinking as that seen in other food cultures. Since the   limitation in the red wine intake, but not this beverage by itself is   included in the Mediterranean dietary pattern, the composition of the red   wine cannot be considered one of the major mechanisms underlying the benefits   of the Mediterranean diet. We introduced in our paper some supplementary   information on this matter.

Round 2

Reviewer 3 Report

I have worried this manuscript to be published because of latent ambiguity; firstly, defining the Mediterranean diet as a solution of NCD, and secondly, defining modified Mediterranean diet as an application to other countries. However, to some extent, the new title has allayed my concern; From “Mediterranean diet: a solution for non-communicable diseases provided by an endangered food culture” To ”The Mediterranean diet: from an environment-driven food culture to an emerging medical prescription”. Moreover, the concepts of Mediterranean dietary pattern and modified Mediterranean diet were properly supplemented as “a whole” and “food synergy”, and author answered the modified Med diets were clarified by Mediterranean Diet Scoring Method. However, you remind that method have to be proved with validation and replication not only in Mediterranean Sea countries but in other countries.